

# Development of the breastfeeding support scale to measure breastfeeding support from lay and professional persons, and its predictive validity in Japan

Keiko Nanishi[1], Joseph Green[1] and Hiroko Hongo[2]

[1] Office of International Academic Affairs, Graduate School of Medicine, The University of Tokyo, Tokyo, Japan
[2] Department of Community and Global Health, Graduate School of Medicine, The University of Tokyo, Tokyo, Japan

## ABSTRACT

**Background:** International and national organizations recommend exclusive breastfeeding for the first 6 months of life, but many women stop earlier. Lay and professional persons can support mothers' efforts to overcome breastfeeding difficulties. Considering breastfeeding support to comprise emotional support, practical help, and information offered to women who desire to breastfeed (by professionals, family members, and others), we developed and tested a scale to measure it in Japan.

**Methods:** A total of 31 items were generated by literature review and from the authors' clinical experiences. Those items were tested with 243 mothers who visited public health centers in Tokyo for their infant's health check-up 3 months after birth. Breastfeeding support and infant feeding status were then assessed 5 months after birth. All the data were collected by using self-administered questionnaires.

**Results:** On the basis of the results of factor analysis, the number of items was reduced to 11. There were three factors: support from breastfeeding peers and from people in specifically named healthcare professions, practical help, and support from people the mother can rely on to help meet emotional needs and address breastfeeding concerns. Internal-consistency reliability (alpha) of scores on the 11-item scale was 0.83 when measured 3 months postpartum and 0.85 when measured 5 months postpartum. Higher scores on the 11-item scale 3 months postpartum were associated with more breastfeeding exclusivity both at that time (Kruskal–Wallis test, chi-squared = 14.871, df = 3, $n$ = 211, $p$ = 0.002, eta-squared = 0.071) and also 5 months postpartum (Kruskal–Wallis test, chi-squared = 8.556, df = 3, $n$ = 159, $p$ = 0.036, eta-squared = 0.054). Further, the area under the Receiver Operating Characteristic curve was 0.73 (95% CI [0.57–0.88]), which indicates that scores on the 11-item scale 3 months postpartum may be useful to predict which mothers will be less exclusive in breastfeeding 5 months postpartum. In conclusion, scores on this 11-item scale were reasonably reliable and valid for measuring breastfeeding support provided by lay and professional persons to mothers in Japan. Further research is required to evaluate this scale's applicability in other settings.

Corresponding author
Keiko Nanishi,
keiko50@m.u-tokyo.ac.jp

# INTRODUCTION

International and national organizations recommend exclusive breastfeeding for the first 6 months of life (*World Health Organization, 2003*). Nonetheless, many mothers who intend to breastfeed for 6 months or longer in fact stop earlier (*Unicef, 2019*). Early cessation can have various causes. Among them are perceived insufficiency of breast milk (*Balogun et al., 2015*; *Dennis, 2002*), difficulty integrating breastfeeding with work outside the home and with other aspects of life (*Balogun et al., 2015*; *Sayres & Visentin, 2018*), and social circumstances that are not conducive to breastfeeding (*Brown, 2017*; *Robinson, Fial & Hanson, 2019*). In 2019, only 42% of infants under 6 months were exclusively breastfed globally (*Unicef, 2019*).

Lay persons and medical professionals can help mothers overcome those challenges (*Hannula, Kaunonen & Tarkka, 2008*; *McFadden et al., 2017*). In Brazil, regular home visits by trained community health workers increased the rate of exclusive breastfeeding 4 months postpartum (*Leite et al., 2005*) and 6 months postpartum (*Coutinho et al., 2005*). A study in Canada showed that telephone-based support from trained peers was effective in maintaining breastfeeding to 3 months postpartum and in improving satisfaction with the infant-feeding experience (*Dennis et al., 2002*). An intervention study in a hospital serving a large population of low-income Latinas found that support from trained peers decreased the rate of early cessation of breastfeeding within 3 months postpartum. A recent systematic review concluded that providing women with additional organized support helps them breastfeed their babies longer (*McFadden et al., 2017*). Also, studies in Sweden and elsewhere showed that women often value emotional support for breastfeeding and practical help for child care from a partner, family members, or others they rely on (*Brown, Raynor & Lee, 2011*; *Cato et al., 2020*; *Juengst et al., 2019*; *Schmied et al., 2011*). Therefore, the availability of support is a key to the continuation of breastfeeding, and tools for measuring it can be useful in identifying needs and in evaluating the effectiveness of interventions.

Several scales for measuring breastfeeding support are available. The Hughes Breastfeeding Support Scale (HBSS) was developed more than 35 years ago in the United States (*Hughes, 1984*). However, since then it has not been widely used and there are very few results of validation testing. *Matich & Sims (1992)* described a scale for measuring breastfeeding support during pregnancy and 3-to-4 weeks postpartum in the United States, although its applicability up to 6 months postpartum remained unclear. Scales have been developed more recently for measuring breastfeeding support among adolescents (*Grassley, Spencer & Bryson, 2013*), and working mothers (*Bai, Peng & Fly, 2008*; *Greene & Olson, 2008*) in the United States, and for mothers in Uganda (*Boateng et al., 2018*).

The support needed to continue breastfeeding may differ according to social context (*McFadden et al., 2017*). We considered that the existing scales may not be appropriate for
measuring breastfeeding support among the majority of mothers in Japan. In Japan, only 0.9% of newborns are born to adolescent mothers (*Ministry of Health, Labour and Welfare, 2019*), and the majority of women who have a baby below 6 months of age are not in the workforce (54% had already resigned from their job and 94% of mothers working at a permanent position were on paid child-care leave, according to a recent national survey) (*Ministry of Health, Labour and Welfare, Director-General for Statistics, Information Policy and Policy Evaluation, 2019*). While a majority of women stay at home to take care of her newborns, only 2% of fathers working in a permanent position take child-care leave (*Ministry of Health, Labour and Welfare, 2019*; *Ministry of Health, Labour and Welfare, Director-General for Statistics, Information Policy and Policy Evaluation, 2019*), which suggests that there is a need to assess breastfeeding support in the context of imbalanced childcare commitment between the parents. More than 93% of pregnant women in Japan intend to breastfeed and a vast majority initiate breastfeeding, but only 55% were found to be predominantly breastfeeding 3 months postpartum (*Equal Employment, Children & Families Bureau, Ministry of Health, Labour & Welfare, 2016*). That gap between original intention and later practice might be due to a lack of support (*Inoue et al., 2012*; *Sriraman & Kellams, 2016*). Appropriate infant feeding support early in the postpartum period is recognized as important by Unicef and the WHO, and compliance with recommendations can be evaluated (*Unicef & World Health Organization, 2018*). However, support to continue breastfeeding, including support from healthcare professionals and from others, has not been quantified in a way that is relevant to Japan. Consequently, in Japan the needs for support among lactating mothers and the effectiveness of breastfeeding support remained unclear. Therefore, we developed a scale to measure breastfeeding support 3 and 5 months postpartum among mothers living in Japan and tested its reliability and predictive validity. In this study, we considered breastfeeding support as comprising emotional support, practical help, and information offered to women who desire to breastfeed by professionals, family members, and others.

## MATERIALS & METHODS

### Item development

Based on the literature on social support and breastfeeding support, the authors conceptualized a framework for the scale. That framework consists of support sources and support type (Table S1). Previous studies in Japan suggested that socio-environmental factors influence breastfeeding. Those factors include appropriate professional support, support from the husband, and the option for mothers to take paid maternity leave for more than 6 months (*Inoue et al., 2012*; *Kaneko et al., 2006*). To reflect the fact that breastfeeding is often influenced by socio-environmental factors, we adopted the ecological model (*Institute of Medicine, 2000*) when considering the source of breastfeeding support. Specifically, we considered that breastfeeding support is provided by social networks that include family, close friends, peers, and health professionals, as well as by mass media and others. In Japan, the marketing of formula milk for babies is not regulated by law, and companies "support" breastfeeding through websites, social networking services, childcare magazines, and face-to-face counseling. Considering the possibility that mothers

recognize them as a source of information to support their breastfeeding, we included companies that produce and sell formula and related goods as a possible source of support. Regarding the types of support, the theoretical concept of social support described by House was also considered (*House, 1981*). Social support is defined as the help provided through social relationships and interactions. House distinguished among four main types of social support: emotional (expressions of empathy, love, trust, and caring), instrumental (tangible aid and services), informational (advice, suggestions, and information), and appraisal (information that is useful for self-evaluation) (*House, 1981*).

Candidate question-items were developed with reference to the literature on social support, breastfeeding support, and factors associated with breastfeeding, and also on the basis of two of the authors' (KN and HH) clinical experience in supporting lactating women. To develop items that reflect support from each support source and type, the first author purposively referred to the literature on factors associated with breastfeeding among women who are healthy, live in high-income countries, and have a single healthy full-term baby (*Dennis, 2002*; *Kaneko et al., 2006*; *Labbok & Taylor, 2008*), and also reviewed other literature on breastfeeding support (*Brown, Raynor & Lee, 2011*; *Burns et al., 2010*; *Hannula, Kaunonen & Tarkka, 2008*; *Hughes, 1984*; *Ito, Fujiwara & Barr, 2013*; *Matich & Sims, 1992*; *Schmied et al., 2011*). Table S2 summarizes the findings of each study reviewed and the items created based on those findings. Reports by Japan's government on the feeding of infants and young children were also reviewed (*Equal Employment Children & Families Bureau, Ministry of Health, Labour & Welfare, 2006*; *Ministry of Health, Labour and Welfare, 2011*). Those reports mentioned that 3-to-5 months postpartum was a common time for mothers to be concerned about infant feeding, but they had no specific information regarding needs for breastfeeding support. Studies on breastfeeding support for working mothers were not included in the literature review because we studied new mothers in the general population, most of whom either were not employed or were on maternity leave.

In addition to the question-items derived from the literature review, the first and the third authors added candidate items on the basis of their experience in clinical pediatrics (KN) and as International Board-Certified Lactation Consultants (KN and HH). Item 6 "I have received free samples or discount coupons for formula milk" and item 20 "In medical facilities I see posters or logos about formula milk" were developed from the idea that free samples and logos at health facilities and elsewhere might make mothers feel that formula is a normative standard supported by health professionals and others while exclusive breastfeeding is considered an extreme choice. Item 18 "I have heard that I must not take any medicine while breastfeeding" and item 22 "I have heard that there are certain things I should not eat while breastfeeding" were created because the authors notice that lactating mothers are often advised to avoid all medications and to avoid specific foods, and that this misinformation is provided both by health professionals and by others. Item 28 "When I leave my baby with other people, I think it will be problematic for them if my baby does not accept a bottle of formula milk" was developed because the authors noticed that mothers often use bottles of formula milk in the belief that doing so reduces the burden on other caregivers, a belief that might reflect inappropriate or insufficient breastfeeding support.
The items were developed in Japanese. A total of 31 items were generated to provide adequate redundancy within each of the four types of support listed above. Some items were positively worded and others were negatively worded. Response choices used a 5-point Likert-type scale from 1 (not agree) through 5 (agree). Those 31 items can be found in Table S3, with English translations.

## Content validation testing

As recommended by Fitzner to improve item validity, the items were reviewed by a panel of experts to assess whether they adequately covered each of the concepts to be measured (*Fitzner, 2007*). The panel comprised five specialists, all of whom were university professors in nursing or midwifery who had clinical experience supporting breastfeeding mothers. Content validity was tested quantitatively by using the Content Validity Index (CVI). CVIs can be computed for each item in a scale (item-level CVI) as well as for the scale as a whole (scale CVI) (*Polit, Beck & Owen, 2007*). To calculate item-level CVIs, each member of the panel of experts rated each item independently on a 4-point Likert-type scale, with higher ratings indicating greater relevance. For each item, the item-level CVI was computed as the number of experts who gave a rating of either 3 or 4, divided by 5 (the total number of experts). The scale-level CVI was assessed by two methods: the proportion of items that was rated either 3 or 4 by all five experts (universal agreement method), and the average of all item-level CVIs (averaging method).

Item-level CVIs for each item are shown in Table S4. The scale-level CVI for the 31 items was 0.94 by the universal agreement method, and 0.99 by the averaging method, both of which are above the recommended minimum of 0.90 (*Polit, Beck & Owen, 2007*). Two items for which the CVI was 0.80 were modified after discussion with the expert panel, and were kept for further analysis.

In addition to being asked to rate the items for the CVIs, the experts were also asked to give comments and suggestions regarding clarity and readability of each item. Those comments and suggestions led to minor editorial revisions.

## Pilot testing

The items were then pilot-tested about a month prior to the main survey with a convenient sample of 27 healthy mothers who visited a public health center in Tokyo's Adachi Ward for their infant's 3-month health check-up. After the participants completed the instrument, which took each mother about 5 min, the first author interviewed them individually to evaluate understandability, face validity, and the instrument's format. Based on feedback from the first 14 mothers, the wordings of eight items were slightly modified. Then 13 other mothers completed the revised scale, and they found all the items to be clear. None of the mothers noted any problem with the format.

## Design and participants

This was a longitudinal study with a survey 3 months after birth and a follow-up survey 2 months later. After the pilot study, we recruited participants for the main study.
Mothers who visited any one of four health centers in Adachi Ward from October to December, 2014 for their infant's 3-month health check-up were invited to participate in the study, if they were at least 18 years old, had a singleton infant, and were fluent in Japanese. Adachi Ward is one of the 23 wards of central Tokyo, and is primarily a residential area. The healthy life expectancy of women in Adachi Ward was the shortest among the 23 wards in Tokyo in 2016 (*Toukyouto hukushi hoken kyoku, 2019*) and it has no Baby-Friendly certified hospital. Adachi Ward had five health centers but one was not included in this study because it did not have enough space for conducting the survey. Among 414 mothers who were approached, 376 (91%) consented to participate in the study. Mothers were excluded if they or their infants had a medical condition that could significantly interfere with breastfeeding. Those conditions included preterm birth ($n = 18$), low birth weight ($n = 22$), macrosomia ($n = 3$), congenital abnormality of the infant ($n = 9$), and a history of admission to an NICU ($n = 3$). In addition, 20 mothers were excluded because they reported that during pregnancy they had intended formula feeding, and 7 mothers were excluded because they visited the health center earlier or later than expected (their infants were younger than 3 months or older than 5 months). Some mothers met more than one exclusion criterion. After the exclusion criteria described above were applied, 243 mothers participated to the study.

All the data were collected by using self-administered questionnaires. The first survey was conducted when the participants were waiting for their appointment at a health center for their infants' 3-month health checkup. The follow-up survey was conducted by postal mail. Among the 243 mothers who completed the first survey, 177 (73%) returned the follow-up survey.

## Psychometric testing

Psychometric testing included factor analysis (described in more detail below) and computation of internal-consistency reliability (coefficient alpha). For validation tests, we hypothesized that higher scores on the breastfeeding social support scale, indicating more support, would be associated with better family functioning and with better infant-feeding status at 3 and 5 months. Family functioning and infant-feeding status were measured as described below.

## Family Apgar

As a construct-validation test, we measured family functioning and computed its correlation with scores on the new breastfeeding social support scale. The Japanese version of the Family Apgar scale was used to measure family functioning. It comprises five question-items regarding family adaptation, partnership, growth, affection, and resolve. Responses were on a 3-point (0, 1, and 2) Likert-type scale, yielding minimum and maximum total scores of 0 and 10. Higher total scores indicate better family functioning. Internal-consistency reliability (coefficient alpha) for the Family Apgar scale was 0.77 in this study.

## Infant-feeding status

Information on infant-feeding status 3 months and 5 months after birth came from the mothers' reports. Mothers were asked which of the following six options best described their infant-feeding method over the previous 24 h: (1) full breastfeeding (exclusive and almost exclusive breastfeeding, which means that no formula milk is given), (2) high partial breastfeeding (breastfeeding for more than 80% of all feedings), (3) medium partial breastfeeding (breastfeeding for 20–80% of all feedings), (4) low partial breastfeeding (breastfeeding for less than 20% of all feedings), (5) token breastfeeding (occasional breastfeeding, not for nutritive purposes), and (6) formula feeding (formula feeding only) (*Labbok & Krasovec, 1990*; *Labbok & Coffin, 1997*).

## Analysis

Factor analysis was done to determine the factor structure and to identify items that contribute to the factors. After the structure and the items to be included in the scale were identified, the reliability and the validity of the scale were tested. To assess reliability, coefficient alpha was computed for the scores on the scale as a whole and for the scores on each subscale. Regarding validity, three hypotheses were tested. First, the total score on the scale was hypothesized to correlate (Pearson's correlation coefficient) with the score on the Family Apgar scale. Second, the total score on the scale 3 months postpartum was hypothesized to be associated with the infant feeding status at that time, and also with the infant feeding status 5 months postpartum. The Kruskal–Wallis test was used to test that hypothesis. That test was used because it is a non-parametric one-way analysis of variance (ANOVA). It uses the ranks of the data rather than their absolute values. This has the potential advantage of not assuming that residuals are normally distributed, so it may be applicable even when the data do not meet that condition. Because it tests for equal medians among all groups, the results of the Kruskal–Wallis test are interpreted in a way that is analogous to the way in which the results of the more commonly used parametric ANOVA are interpreted. The difference is that with the Kruskal–Wallis test the interpretation refers to medians whereas with parametric ANOVA the interpretation refers to means. For example, a $p$ value less than 0.05 would indicate that the median of the population of one or more of the groups is not the same as the median of the population of one or more of the other groups. The effect size (eta-squared) was computed as chi-squared$/(n - 1)$. Finally, the area under the Receiver Operating Characteristic curve was calculated to assess the degree to which the total score and each subscale score 3 months postpartum could be used to predict which mothers would be in a low category of infant-feeding status 5 months postpartum.

## Ethical considerations

Ethical approval was obtained from the Research Ethics Committee of the Graduate School of Medicine at the University of Tokyo (Ethical Application Ref: 10620), and written informed consent was received from all participants.

## RESULTS

### Characteristics of participants and infant-feeding status

Characteristics of participants are shown in Table 1. Their mean age was 31.7 (SD 5.0) years. Among the 243 participants, 197 (81.1%) had a vaginal delivery, 144 (59.3%) were primiparas, 157 (65.6 %) indicated that during pregnancy they had intended to breastfeed exclusively, and 84 (34.6%) had previous experience of any breastfeeding for more than 5 months. Fourteen mothers (5.9%) were working at 3 months postpartum and an additional 12 (4.9%) had a plan to return to work outside the home before 6 months postpartum. Compared with those who returned the follow-up survey, those who did not return it were younger, less educated, and more likely to be single mothers. However, there was no significant difference between the two groups in mode of delivery, intention to breastfeed, previous experience breastfeeding, parity, country in which the participant was raised, working status, or financial status.

Table 2 shows infant feeding status at the time of the survey. Three months postpartum, 137 (56.4%) reported full breastfeeding within the 24 h before the survey. Five months postpartum, that number was 109 (61.6% of those who responded). Not many were token feeding (4 (1.7%) at 3 months postpartum and 2 (1.1%) at 5 months postpartum) or completely formula feeding (22 (9.1%) at 3 months postpartum and 14 (7.9%) at 5 months postpartum). Among the 159 women who reported their feeding method both 3 months and 5 months postpartum, during the interval between those two reports 7 women increased the exclusivity of breastfeeding, while 22 either reduced the exclusivity of breastfeeding ($n = 18$) or stopped breastfeeding ($n = 4$).

### Initial psychometric testing, and modifications based on those results

The mean and SD of each of the 31 items' scores is shown in Table 3. The scores on items 16 and 27 had particularly low SDs.

The scree plot resulting from factor analysis of the 31 items (Fig. 1) clearly had two sections, which were separated by an "elbow" at factor number 4. Looking at the curve in Fig. 1 from left to right, the eigenvalues decrease very "steeply" between factor number 1 and factor number 3, but much more gradually starting at factor number 4. This indicated that the number of factors to be retained was three (Fig. 1). That is, much of the variability in the responses to the 31 questions could be explained by variability in only three underlying factors. This was not unexpected, given that the 31 questions were intended to ask about a smaller number of separate sources and types of breastfeeding support. Figure 1 was interpreted as meaning that the 31 questions were measuring three underlying variables. Next, with the number of factors restricted to three, after varimax rotation the items with factor loadings less than 0.4 were deleted: Those were items 6, 10, 11, 12, 13, 14, 15, 16, 18, 19, 20, 22, 23, 24, 25, 27, 28, and 30. With the remaining 13 items, the same factor analysis procedure was done again and items 21 and 29 were omitted. By that process, the only items retained were those that were best for measuring the three underlying factors.

**Table 1 Characteristics of participants.** The numbers of mothers and the percentages of those who responded are shown. The data are stratified by participation status: mothers who completed the first survey 3 months postpartum, mothers who returned the follow-up survey 5 months postpartum, and those who were lost to follow-up. The characteristics of those who returned the follow-up questionnaire were compared with the characteristics of those who did not, and the p-values are shown.

| | Completed the survey 3 months postpartum (n = 243) | Completed the survey 5 months postpartum (n = 177) | Dropped out from the follow-up (n = 66) | P value[a] |
|---|---|---|---|---|
| Age in years (mean, SD) | 31.7 (5.0) | 32.2 (4.4) | 30.2 (6.2) | 0.016 |
| Country in which participant was raised | | | | |
| Japan | 233 (95.9%) | 172 (97.2%) | 61 (92.4%) | 0.140 |
| Others | 10 (4.1%) | 5 (2.8%) | 5 (7.6%) | |
| Highest level of schooling | | | | |
| Junior high | 11 (4.5%) | 4 (2.3%) | 7 (10.8%) | 0.002 |
| High school | 53 (21.8%) | 34 (19.2%) | 19 (29.2%) | |
| College[b] or equivalent | 88 (36.2%) | 64 (36.2%) | 24 (36.9%) | |
| University[c] or higher | 90 (37.0%) | 75 (42.4%) | 15 (23.1%) | |
| Financial status | | | | |
| No financial worries | 38 (15.6%) | 25 (14.1%) | 13 (19.7%) | 0.138 |
| Not very worried | 84 (34.6%) | 68 (38.4%) | 16 (24.2%) | |
| Somewhat worried | 89 (36.6%) | 64 (36.2%) | 25 (37.9%) | |
| Worried | 32 (13.2%) | 20 (11.3%) | 12 (18.2%) | |
| Marital Status | | | | |
| Married or having a steady partner | 234 (96.3%) | 173 (97.7%) | 61 (92.4%) | 0.037 |
| Delivery mode | | | | |
| Vaginal delivery | 197 (81.1%) | 143 (81.7%) | 54 (83.1%) | 0.807 |
| Caesarian section | 43 (17.7%) | 32 (18.3%) | 11 (16.9%) | |
| Intention to breastfeed[d] | | | | |
| Exclusive breastfeeding | 157 (65.6%) | 112 (63.3 %) | 45 (68.2%) | 0.477 |
| Partial breastfeeding | 86 (35.4%) | 65 (36.7%) | 21 (31.8%) | |
| Primipara | 144 (59.3%) | 105 (59.3%) | 39 (59.1%) | 0.424 |
| Previous experience of breastfeeding a baby for more than 5 months | 84 (34.9%) | 60 (34.3%) | 24 (36.4%) | 0.763 |
| Working status | | | | |
| Working at 3 months postpartum | 14 (5.9%) | 6 (3.4%) | 8 (12.7%) | 0.094 |
| Planning to return to work before 6 months postpartum | 12 (4.9%) | 9 (5.1%) | 3 (4.8%) | |
| Planning to return to work after 6 months postpartum | 108 (45.2%) | 83 (47.2%) | 25 (39.7%) | |
| No plan to work | 105 (43.2%) | 78 (44.3%) | 27 (42.9%) | |

**Notes:**
[a] Comparison between those who returned and did not return the follow-up questionnaire.
[b] Typically a 2-year course after high school.
[c] Typically a 4-year course after high school.
[d] Those who intended formula feeding were excluded from the analysis.

The resulting scale had 11 items, and factor analysis with those 11 items confirmed that each of them had a loading greater than 0.4 on one, and only one, of the three factors (Table 4). Items 7, 8, 9, and 31 loaded strongly on the first factor. Because of the content

**Table 2 Infant feeding status 3 months and 5 months postpartum.** The numbers of mothers and the percentages of those who responded are shown, by infant feeding status 3 months postpartum and 5 months postpartum. Infant feeding status is shown in six categories.

| Infant feeding status at the time of the survey | 3 months ($n = 242$) | 5 months ($n = 177$) |
|---|---|---|
| Full breastfeeding (exclusive and almost exclusive) | 137 (56.6%) | 109 (61.6%) |
| High partial breastfeeding (breastfeeding for more than 80% of all feedings) | 49 (20.2%) | 23 (13.0%) |
| Middle partial breastfeeding (breastfeeding for 20–80% of all feedings) | 24 (9.9%) | 23 (13.0%) |
| Low partial breastfeeding (breastfeeding for less than 20% of all feedings) | 6 (2.5%) | 6 (3.4%) |
| Token feeding (occasional breastfeeding, not for nutritive purposes) | 4 (1.7%) | 2 (1.1%) |
| Formula feeding (only formula feeding only) | 22 (9.1%) | 14 (7.9%) |

that those four items had in common, they were considered to comprise a subscale measuring support from breastfeeding peers and from people in specifically named healthcare professions. Items 2, 5, and 26 loaded strongly on the second factor. Because of the content that those three items had in common, they were considered to comprise a subscale measuring practical help. Similarly, items 1, 3, 4, and 17 loaded strongly on the third factor, and they were considered to comprise a subscale measuring support from people the mother can rely on to help meet emotional needs and address breastfeeding concerns.

## Reliability testing

Three months postpartum, coefficient alpha of the scores on the 11-item scale was 0.83, and 5 months postpartum it was 0.85. Three months postpartum, the alphas for the scores on the subscales were 0.78 (breastfeeding peers and named professions), 0.80 (practical help), and 0.81 (support to meet emotional needs). Five months postpartum, the alphas for the scores on the subscales were 0.79 (breastfeeding peers and named professions), 0.86 (practical help), and 0.81 (support to meet emotional needs).

## Validation testing

We hypothesized that mothers in well-functioning families would have higher scores on the 11-item breastfeeding support scale. That hypothesis was supported by the positive correlation between the 11-item scale score and the Family Apgar score 3 months postpartum ($r = 0.47$, $p < 0.001$).

Table 5 shows the mean scale scores (11 items) of women in each category of infant-feeding status 3 and 5 months postpartum. Among the mothers who completed the scale 3 months postpartum, very few were token feeding ($n = 2$) or completely formula feeding ($n = 3$) 5 months postpartum, so those two categories were combined with the

| Item number | Item[a] | Mean[b] | Standard deviation |
|---|---|---|---|
| 1 | There is someone with whom I can easily and openly discuss breastfeeding | 4.30 | 1.12 |
| 2 | There is someone who helps with other child care and with housework such that it's easy for me to make time to breastfeed my baby | 3.31 | 1.46 |
| 3 | There is someone close to you who gives you emotional support in breastfeeding | 3.89 | 1.24 |
| 4 | There is someone who tells me about positive experiences of breastfeeding | 3.47 | 1.45 |
| 5 | There is someone who helps you with other child care and with housework such that it's easy for you to take care of your baby | 3.68 | 1.39 |
| 6[c] | I have received free samples or discount coupons for formula milk | 2.48 | 1.65 |
| 7 | If necessary, there is someone other than family or friends (e.g. health care provider, breastfeeding support group member) whom I can consult on breastfeeding | 3.69 | 1.48 |
| 8 | Most health care providers (doctors, public health nurses, midwives, etc.) support you in breastfeeding | 3.94 | 1.08 |
| 9 | Health care providers including doctors, public health nurses, or midwives tell me about the benefits of breastfeeding | 3.99 | 1.12 |
| 10 | I can breastfeed comfortably when I'm out and about | 3.47 | 1.33 |
| 11[c] | I see formula milk with product information saying that breastmilk and artificial milk do not differ much in their health benefits for babies | 2.22 | 1.25 |
| 12[c] | I have received advice regarding infant feeding from people employed by the dairy industry ("advisors", nutritionists, etc.) | 3.44 | 1.47 |
| 13 | Information from books, magazines, and the Internet is useful for breastfeeding | 3.86 | 1.09 |
| 14[c] | Sometimes I provide something other than breastmilk to my baby because housework or parenting of an elder child gets in the way | 3.80 | 1.53 |
| 15[c] | Information from television, newspapers, etc. sometimes makes me anxious about breastfeeding | 4.25 | 1.14 |
| 16[c] | There is someone close to me who encourages me to wean my baby from the breast soon | 4.55 | 0.94 |
| 17 | There is someone I can talk with whenever I have issues with breastfeeding | 3.95 | 1.25 |
| 18[c] | I have heard that I must not take any medicine while breastfeeding | 3.42 | 1.47 |
| 19[c] | There are discrepancies among what health care providers (doctors, public health nurses, midwives, etc.) say about breastfeeding | 2.76 | 1.28 |
| 20[c] | In medical facilities I see posters or logos about formula milk | 2.78 | 1.51 |
| 21[c] | Information on breastfeeding from mass media or the Internet confuses me | 3.32 | 1.39 |
| 22[c] | I have heard that there are certain things I should not eat while breastfeeding | 2.07 | 1.21 |
| 23[c] | There is someone close to me who encourages me to provide something other than breastmilk to my baby | 3.75 | 1.44 |
| 24[c] | I have trouble finding places to breastfeed when I'm out and about | 2.30 | 1.24 |
| 25[c] | According to product information regarding formula milk, giving formula milk has health benefits for babies | 2.57 | 1.12 |
| 26 | There are people around you who help you get enough rest | 3.67 | 1.22 |
| 27 | I can breastfeed my baby comfortably at home | 4.71 | 0.62 |
| 28[c] | When I leave my baby with other people, I think it will be problematic for them if my baby does not accept a bottle of formula milk | 1.78 | 1.15 |
| 29[c] | I feel pressured to breastfeed my baby | 3.70 | 1.41 |
| 30 | There are services from the city or other local government that help me take care of my baby | 3.60 | 1.10 |
| 31 | If necessary, I can consult with health care providers (doctors, public health nurses, midwives, etc.) on how to breastfeed | 3.79 | 1.16 |

**Notes:**
[a] Each item was developed and presented in Japanese. The text in the table was translated into English by the authors. The English translation provided here is to be used not for collecting data, but rather for informational purposes only. It has not been tested in English for use among English-speaking mothers. For example, in the English translation provided here, some items contain "I" while others contain "you". Neither the developers nor the users found the mix of the corresponding Japanese expressions to be unacceptable, but the wording might be standardized to use only "I" or only "you" if the instrument were to be tested in English and used in English.
[b] Each item's score can range from 1 to 5.
[c] These items were negatively worded. Their scores were reversed, such that higher scores indicate more support.

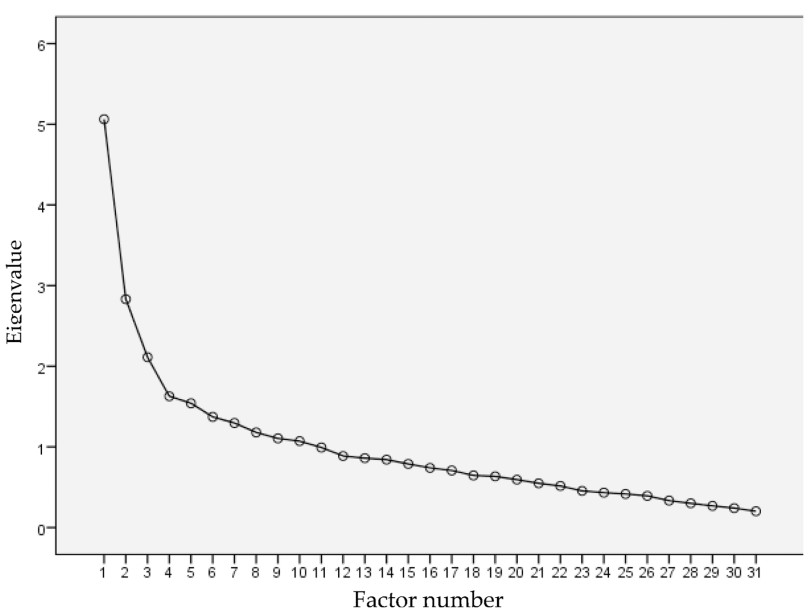

**Figure 1 Scree plot from factor analysis with the original 31 items.** The scree plot from factor analysis with the original 31 items is shown. Because the "elbow" in the curve was at factor number 4, three factors were retained.                                                               

**Table 4 Factor loadings of 11 items after varimax rotation.** The factor loadings of the 11 items after varimax rotation are shown.

| Item number | Item content[a] | Factor 1[b] | Factor 2[c] | Factor 3[d] |
|---|---|---|---|---|
| 7 | If necessary, there is someone other than family or friends (e.g. health care provider, breastfeeding support group member) whom I can consult on breastfeeding | **0.576** | 0.072 | 0.244 |
| 8 | Most health care providers (doctors, public health nurses, midwives, etc.) support you in breastfeeding | **0.842** | −0.011 | 0.097 |
| 9 | Health care providers including doctors, public health nurses, or midwives tell me about the benefits of breastfeeding | **0.730** | 0.079 | 0.133 |
| 31 | If necessary, I can consult with health care providers (doctors, public health nurses, midwives, etc.) on how to breastfeed | **0.586** | 0.023 | 0.199 |
| 2 | There is someone who helps with other child care and with housework such that it's easy for me to make time to breastfeed my baby | −0.003 | **0.831** | 0.192 |
| 5 | There is someone who helps you with other child care and with housework such that it's easy for you to take care of your baby | 0.029 | **0.855** | 0.139 |
| 26 | There are people around you who help you get enough rest | 0.093 | **0.501** | 0.279 |
| 1 | There is someone with whom I can easily and openly discuss breastfeeding | 0.120 | 0.192 | **0.776** |
| 3 | There is someone close to you who gives you emotional support in breastfeeding | 0.193 | 0.378 | **0.571** |
| 4 | There is someone who tells me about positive experiences of breastfeeding | 0.353 | 0.261 | **0.507** |
| 17 | There is someone I can talk with whenever I have issues with breastfeeding | 0.354 | 0.162 | **0.698** |

**Notes:**
[a] English translation of the questions asked.
[b] Considered to comprise a subscale measuring support from breastfeeding peers and from people in specifically named professionals.
[c] Considered to comprise a subscale measuring practical help.
[d] Considered to comprise a subscale measuring support from people the mother can rely on to help meet emotional needs and address breastfeeding concerns.
Factor loadings greater than 0.4 are shown in bold.

**Table 5 Scores on the 11-item scale, measured 3 months postpartum, by infant-feeding status 3 months and 5 months postpartum.** Means of total scores measured 3 months postpartum are shown, by infant feeding status. The scores 3 months postpartum are shown in the upper lines and the scores 5 months postpartum are shown in the lower lines.

| Infant-feeding status | n (%[a]) | Total score mean (SD) |
|---|---|---|
| 3 months postpartum | | |
| Full breastfeeding | 132 (62.6) | 42.6 (8.7) |
| High-partial breastfeeding | 46 (21.8) | 41.9 (6.9) |
| Medium-partial breastfeeding | 21 (10.0) | 37.8 (8.0) |
| Low-partial breastfeeding, token breastfeeding, or formula feeding | 12 (5.7) | 33.3 (10.4) |
| 5 months postpartum | | |
| Full breastfeeding | 104 (61.6) | 41.9 (8.9) |
| High-partial breastfeeding | 22 (13.0) | 43.6 (7.1) |
| Medium-partial breastfeeding | 22 (13.0) | 40.3 (6.3) |
| Low-partial breastfeeding, token breastfeeding, or formula feeding | 11 (12.4) | 35.4 (8.4) |

**Note:**
[a] Among those who completed the scale 3 months postpartum and also reported feeding status at the time of the survey.

low-partial breastfeeding category ($n = 6$) to make a group of 11 mothers. Higher scores on the 11-item scale 3 months postpartum were associated with more breastfeeding exclusivity both at that time and also 2 months later (i.e. both 3 months and also 5 months postpartum). Specifically, the results of the Kruskal–Wallis test 3 months postpartum were chi-squared = 14.871 (df = 3, $n = 211$), $p = 0.002$, eta-squared = 0.071, and the results of the same test 5 months postpartum were chi-squared = 8.556 (df = 3, $n = 159$), $p = 0.036$, eta-squared = 0.054.

The area under the Receiver Operating Characteristic curve (AUC) was 0.73 (95% CI [0.57–0.88]), which indicates that scores on the 11-item scale 3 months postpartum may be useful to predict which mothers will be in a low category of infant-feeding status (i.e., low-partial breastfeeding, token breastfeeding, or formula feeding) 5 months postpartum. Regarding the subscales, the AUC was 0.78 (95% CI [0.63–0.94]) for the subscale measuring support from breastfeeding peers and from people in specifically named healthcare professions, 0.44 (95% CI [0.27–0.94]) for the subscale measuring practical help, and 0.74 (95% CI [0.60–0.88]) for the subscale measuring support from people the mother can rely on to help meet emotional needs and address breastfeeding concerns.

## DISCUSSION

The purpose of the study was to develop an instrument for measuring support that might help mothers continue breastfeeding. After the initial psychometric testing and modifications based on the results of those tests, the final version of the Breastfeeding Support Scale had 11 items with a three-factor structure: support from breastfeeding peers and from people in specifically named healthcare professions, practical help, and support from people the mother can rely on to help meet emotional needs and address

breastfeeding concerns. The scores were reasonably reliable and valid for measuring breastfeeding support.

The structure of the scale is consistent with previous studies about breastfeeding support in developed countries (*Emmott & Mace, 2015*; *Emmott, Page & Myers, 2020*; *Fox, McMullen & Newburn, 2015*; *Negron et al., 2013*; *Schmied et al., 2011*). Those studies indicated that the effectiveness of breastfeeding support is a complex function of the provider of support and the type of support. *Emmott, Page & Myers (2020)* found that mothers who received support from a wide network of people including family, friends, health professionals, and trained peer supporters breastfed longer than those who did not receive much support or who received only family-based support. Also, previous studies suggested that breastfeeding support includes informational support (*Emmott & Mace, 2015*), emotional support (*Fox, McMullen & Newburn, 2015*; *Negron et al., 2013*; *Schmied et al., 2011*), and practical support (*Emmott & Mace, 2015*; *Negron et al., 2013*). The Breastfeeding Support Scale has three factors, with each factor reflecting both a source and a type of support. The first factor covers informational support and appraisal from breastfeeding peers and from people in specifically named healthcare professions. The second factor covers practical support that gives mothers enough time for breastfeeding, childcare, and rest. The third factor reflects having somebody who responds to mothers' emotional needs to continue breastfeeding.

When we developed the candidate items for the scale, we tried to cover a wide range of potential sources of support: media, infant formula companies, and the social-cultural environment. Some of those candidate items mentioned informational support from media (i.e., "Information from books, magazines, and the Internet is useful for breastfeeding."), from infant formula companies (i.e., "I see formula milk with product information saying that breastmilk and artificial milk do not differ much in their health benefits for babies.", which is reverse scored), and from a breastfeeding-friendly environment (i.e., "I can breastfeed comfortably when I'm out and about."). None of the candidate items that asked about support from media, infant formula companies, and the social-cultural environment remained after the factor analysis. This indicates that, among these mothers, positive and negative influences from media, infant formula companies, and the social-cultural environment are not strongly associated with support from family, friends, health professionals, and trained peer supporters.

Psychometric testing of the Breastfeeding Support Scale among mothers in Japan indicated that it had reasonable reliability and validity to measure breastfeeding support both 3 and 5 months postpartum. At both times, the values of alpha for the overall scale and for each of the three subscales were all substantially higher than 0.7, which is often used as a minimum for group-level comparisons. The results of all three validation tests were as hypothesized. First, the total score on the Breastfeeding Support Scale was correlated with the score on the Family Apgar scale, which measured family functioning. Second, the total score on the Breastfeeding Support Scale was associated with infant feeding status 3 months and 5 months postpartum. Finally, the area under the Receiver Operating Characteristic curve indicated that the total score of the Breastfeeding Support

Scale could be used to predict which mothers would be in a low category of infant-feeding status 5 months postpartum.

Consistent with previous studies (*Emmott, Page & Myers, 2020*), the results of the present study suggest that different types of support from different sources might have different effects on breastfeeding. The AUCs of each subscale suggested that peer and professional support, and having somebody who responded to emotional needs, are the keys to higher breastfeeding exclusivity. In contrast, practical help, including having someone who helps with housework and childrearing, did not clearly predict breastfeeding 2 months later. That might suggest such practical help does not necessarily increase breastfeeding exclusivity as previously indicated in studies in Japan (*Ito, Fujiwara & Barr, 2013*) and the UK (*Emmott & Mace, 2015*). A possible explanation might be fathers' insufficient knowledge and skills regarding breastfeeding support. Another reason might be the significant gender gap in Japan in unpaid housework. When fathers become more involved in housework and childrearing, practical help, which is usually offered by a partner, may impact breastfeeding outcomes. However, the wide range of the confidence interval of the AUC for the practical help subscale suggests that more information may be required before that finding can be interpreted clearly.

There are several limitations to this study. The study was conducted in four public health centers in Tokyo Japan, so generalizations to populations in other areas should proceed only with caution. Mothers younger than 18 years old were not included and the majority of the participants were on paid leave or were not working outside their home, so further testing may be needed before the scale is used with adolescents or with working mothers. In addition, after the data were collected Japan's Ministry of Health, Labor and Welfare released a revised version of its guidelines for health professionals on the feeding of infants and young children (*Jyunyuu rinyuu no shien gaido kaitei ni kannsuru kennkyuukai, 2019*). In that revised version, the term "breastfeeding promotion" was omitted and the 54-page document has only two sentences regarding the benefits of breastfeeding. Those guidelines might discourage professional breastfeeding support in Japan. Further testing may be necessary to understand how the present scale performs after that policy change. The literature review to develop the items was also done more than 6 years ago, however, according to our knowledge there has been no literature published in Japan that would suggest a need to revise the scale. Finally, as all the data were collected by self-report, there might be social desirability bias, such as breastfeeding reported as more exclusive or support from family rated better. The English translation of the scale provided here is to be used for informational purposes only, so further psychometric testing is required to assess reliability and validity outside Japan.

## CONCLUSIONS

Initial evidence favors the use of the Breastfeeding Support Scale among mothers in Japan to measure breastfeeding support provided by a wide range of people including peers, family, and professionals. Further research is required to evaluate the scale's applicability in other settings.

## ACKNOWLEDGEMENTS

The authors are grateful to the staff members of the Adachi public health center for their help in conducting the study. We also thank the following researchers for their critical review of the contents of the scale: Dr. Shigemi Iriyama, Dr. Masako Matsunaga, Dr. Megumi Haruna, Dr. Junko Miyazawa, and other professors.

### Funding

The study was funded by a JSPS KAKENHI grant: Grant No.23792640. The funders had no role in study design, data collection and analysis, decision to publish, or preparation of the manuscript.

### Grant Disclosures

The following grant information was disclosed by the authors:
JSPS KAKENHI: 23792640.

### Competing Interests

The authors declare that they have no competing interests.

### Author Contributions

- Keiko Nanishi conceived and designed the experiments, performed the experiments, analyzed the data, prepared figures and/or tables, authored or reviewed drafts of the paper, and approved the final draft.
- Joseph Green analyzed the data, prepared figures and/or tables, authored or reviewed drafts of the paper, and approved the final draft.
- Hiroko Hongo performed the experiments, authored or reviewed drafts of the paper, and approved the final draft.

### Human Ethics

The following information was supplied relating to ethical approvals (i.e., approving body and any reference numbers):

Ethical approval was obtained from the Research Ethics Committee of the Graduate School of Medicine at the University of Tokyo (Ethical Application Ref: 10620).
We received written informed consent from all participants.

### Data Availability

The raw data are available in the Supplemental File.

### Supplemental Information

Supplemental information for this article can be found online at http://dx.doi.org/10.7717/peerj.11779#supplemental-information.

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
