# Peer review of "Development of the breastfeeding support scale to measure breastfeeding support from lay and professional persons, and its predictive validity in Japan"

_PeerJ, doi:10.7717/peerj.11779_

## Round 0.1 · original submission · Minor Revisions

All three referees positively reviewed the manuscript, providing constructive suggestions.

Most criticisms can be addressed by editing.

If the authors assessed participants' socioeconomic status, the inclusion of such information would increase the study's value, as reviewer 1 suggests.

·

Basic reporting

Abstract:
Authors note the scale captures support from lay and professional persons in Japan, but some item questions relate to more institutional or cultural supportive factors rather than support from an individual. I would advise authors to think further about what aspect of support the items capture, to ensure the developed scale is accurately described.

Introduction:
Given the study aim is to develop a relevant measure of breastfeeding support provided to mothers in Japan, I believe authors should elaborate and clarify how support is conceptualised, and how the items relate to these conceptualisations of support in the introduction. While authors mention four types of support and the discussion does provide further details, some of the items are difficult to categorise as support (such as information about formula); rather, they seemed to be capturing broader ecological contexts. This is not a problem, but authors may need to reframe what their items were developed to capture – rather than support itself, perhaps the items relate to a broader measure a breastfeeding-supportive environment? (It might be helpful for authors to think about the measures in relation to the ecological systems theory, which clarifies different “layers” of the environment and how they may impact individuals. Note, this is not a suggestion to include ecological systems theory into the manuscript; just a tool to help authors think about support.) I recommend redevelopment of the introduction which would clarify the purpose and methods of the study. My specific suggestions are as follows, which I hope the authors find helpful:
• In the first paragraph, the scale of the public health issue around breastfeeding duration could be clarified further by providing basic statistics on breastfeeding rates across the globe. Currently the issue is only verbally described, but this should be backed up by evidence.
• In the first paragraph three reasons are given on why many mothers stop breastfeeding earlier, but this are not exclusive. I recommend rephrasing to “because of problems such as” which clarifies that there are many reasons.
• Throughout the introduction, evidence is discussed with assumed universal applicability of the findings. However, the determinants of breastfeeding likely vary between populations, as evidenced from significant variations in breastfeeding initiation rates and average duration length within and between populations. I would recommend authors are more specific about the population where the evidence comes from. E.g., Studies from the US suggest…
• Overall, support is not clearly defined: Authors outline various studies which found associations between various support measures and breastfeeding outcomes, but it is not clear how the different measures of support should be interpreted (i.e., are they all the same thing, or are they different?). Given the focus of the study is around support, I believe the paper requires a few sentences clarifying what support is.
• Line 63-64: The importance of measuring support was not particularly clear in this sentence. Further, promoting breastfeeding can conflict with support, therefore measuring support is not necessarily key for breastfeeding promotion. Better phrasing may be something like, “Therefore, accurately measuring the availability of support for mothers is a key step in identifying their support needs and providing effective breastfeeding support.”
• Lines 70-71: “although quantitative information on support up to 6 months postpartum is also needed” – this is unclear; is this in regards to the scale, or is this a comment about something else? (I.e., needed by whom?).
• Line 72-74: mothers in Uganda is specified, but it is not specified for scales among adolescents and working mothers. Were the scales for adolescents and working mothers therefore proposed to be cross-culturally applicable scales, but the one in Uganda for mothers in Uganda only? I am wondering whether, in fact, the support scale for adolescent mothers and working mothers are also population specific. If so, the population should be specified (i.e., USA). I am not confident we should assume studies in the US would automatically apply to the rest of the world!
• Lines 75-77: Authors note, “Because the support needed to continue breastfeeding may differ according to social context, we considered that the existing scales may not be appropriate for measuring breastfeeding support among the majority of mothers in Japan.” This statement would be better evidenced if cross-cultural variations in how support influences breastfeeding was reviewed above. Up to this point, authors present a rather universal picture of how support measures associate with breastfeeding – but surely, this may vary across populations, not just in Japan?
• Line 78: “Very few” is subjective – are there any national statistics available you could cite?
• Line 79: “Majority” could vary from 51% to 99% - as above, are there any national statistics available you could cite?
• Lines 83-84: “That gap between original intention and later practice might be due to a lack of support.” Many studies have reported the gap between breastfeeding intention/desire and practice and attributed this to the lack of support, so this hypothesis would be better supported by referencing existing studies which have also stated this.
• Lines 86-87: “has not been quantified in a way that is relevant to Japan” requires more explanation. I recommend providing more detail on the existing scales in Lines 65-74, and raising specific issues with existing scales as an example in the final paragraph of the introduction to support this statement.

Other comments on basic reporting:
• Line 107-114: A good definition of support is provided here, but this would be better moved to the introduction where concepts of support are first introduced.
• Line 123: The reference here is unclear – it seems like Fitzner 2007 provides further details on the item review process? Perhaps better phrasing would be “As recommended by Fitzner (2007) to improve item validity, the items were reviewed by…“
• Paragraph one and paragraph two of the methods should be combined as they relate to the same subject. Line 126 could be rephrased as: “Content validity was tested quantitatively by using the Content Validity Index (CVI).” Otherwise I think this section is very clear.
• Line 115-116: This should be clarified in the SI; i.e., link each item to the listed types of support. This will help the reader better interpret the items.
• Pilot testing: Please clarify roughly when and where this was carried out.
• Line 240: “Not many” is subjective – please provide the number.
• For Table 3, I think it would be helpful to include the item questions with the item number, so readers do not have to keep referring back to Table 2, and to include a brief description/label for each factor. This will help readers better understand/assess the factors.
• Lines 281-285: This information is about deriving variables and provides descriptive statistics, and it should be placed in the methods (around Line 240?)

As a Japanese and English speaker, I also have some minor suggestions regarding the English translation of the Japanese items:

Item 1: Better translation may be “There is someone with whom I can speak to about breastfeeding” where kigaruni is implied. “Frankly” suggests some formality and assertiveness, so I am not sure this is a suitably translation.
Item 2: Emotional support may be better term rather than psychological support? As phycology is often linked to the “brain” and emotion to the “heart”
Item 8: Health care providers "such as" doctors, public health nurses, and midwives tell me about the benefits of breastfeeding.
Item 10: I can breastfeed comfortably when "I’m out and about" may be a better translation – as being away from home does not necessarily mean being out (you could be staying with your family at their house, for example).
Item 14: Just a point that this item seems to be a leading question with implicit judgement that not breastfeeding is negative and undesirable. A more neutral/less judgmental phrasing would have been something like “Sometimes I provide something other than breastmilk to my baby, because housework or parenting of an elder child gets in the way.” While this is not included in the final factor analysis, authors may want to mention this in the limitations.
Item 17: There is someone I can talk with whenever I have "issues" with breastfeeding. may be better
Item 18: There are discrepancies among what health care providers "such as" doctors, public health nurses, and midwives say about breastfeeding.
Item 24: I have trouble finding places to breastfeed when "I’m out and about."
Item 31: If necessary, I can consult with health care providers "such as" doctors, public health nurses, and midwives on how to breastfeed.

Experimental design

Overall I thought this was a well-executed study and the methods were clear, but I would have liked some more information on the methods regarding item development. In particular, further detail on the literature review process would help the reader understand the process of item development:

Firstly, what exactly were authors hoping to measure? I ask this, as some items seem to relate to perceived support, while others seem to relate to advertising of formula and it is not clear how this relates to support. (For example, how is an advert on formula related to support? Given that information on formula could be very supportive for mothers who are formula feeding, and an advert is not necessarily devised to be supportive, could this really be conceptualised as support, or is it to do with establishing norms?). Perhaps because the conceptualisations of support is relatively weak in the introduction, it is not clear how the items are intended to measure support.

Second, how were the literature reviewed identified? If all the reviewed papers were cited, it does not look like a comprehensive list of available studies and looks relatively dated. What kind of studies on support for working mothers were excluded? I note that the study was conducted in 2014, which would explain why some more recent papers are not included in the references. This is not an issue, but it would help the reader understand why more recent papers were not included if you provide the timeframe of when these steps were taken. I would recommend providing a more detailed outline of how the literature was identified/reviewed in the supplementary information.

Finally, each item in SI1 could be referenced to papers where it was derived from. Again, this will show the readers that the items are based on evidence & clarify the process of item development further.

Validity of the findings

Were any information on participant socioeconomic position collected, such as education level or household income? As this information will provide further detail on the generalisability of the findings. If not, this should be outlined in the limitations.

I would have liked more information on the Adachi Ward which will help with inference; i.e., how do the population/mothers in the Adachi Ward compare to the rest of Japan?

As mentioned under my comments on basic reporting, I recommend authors clarify the theories and concepts around what was measured so the findings are inferred appropriately.

Additional comments

Thank you for the opportunity to review this manuscript. Overall, I thought the paper was very interesting & the study was well-executed, but I recommend redevelopment of the introduction which would clarify the purpose of the study, what the items are actually measuring, and help the reader infer the study findings appropriately. I have provided a detailed review, and I hope this is useful for the authors in improving the manuscript.

·

Basic reporting

This work is interesting. In order to present the need to develop new tools related to breastfeeding for Japanese women, a wide range of up-to-date literature needs to be reviewed. Especially in the introduction section, the reference should be changed to the latest.

Experimental design

no comment

Validity of the findings

The results in this study are clearly presented. However, the data used in the study was collected six years ago, which may not reflect changes over time. These limitations need to be described by adding them to the discussion as a limitation.

Additional comments

Introduction: Considering the cultural characteristics of Japan, it is necessary to present special matters that affect the support of breastfeeding.
Line 77-80: A reference should be presented in this sentence.
Line 119: S1 Table & Line 135: Table S2 - Please unify the way of the supplemental table numbers are written in the two parts.

Reviewer 3 ·

Basic reporting

1. This is a nicely structured, written and presented manuscript.
2. The literature review cites the most important and relevant literature, which is used to identify the appropriate study design, and the introduction succinctly positions the study in the literature and well articulates the rationale for the study.
3. The four tables are clear and informative. Data availability is indicated, and the file provided.

Experimental design

4. The survey design and statistical analysis is clearly explained, and the factor analysis appropriate for the research question.
5. The statistical analysis appears to be conducted to a high technical standard, and is clearly and succinctly reported. The authors could consider providing a sentence or two explaining the choice of factor analysis and the Kruskal-Wallis test, and an example of how to interpret the statistics, for readers unfamiliar with factor analysis and non parametric tests.
6. The authors could clarify in the abstract that the data is based on interview and maternal self-report of breastfeeding practice.
7. Some information could be provided comparing the characteristics of participants lost to follow up to indicate possible bias in the final sample.

Validity of the findings

8. The authors conclusions appear well supported by the data and analysis, and the extent of generalisability appropriately acknowledged.
9. Limitations could include reference to social desirability bias as a limitation in interpreting survey responses.

Additional comments

10. This is a well conducted and nicely presented study and addresses an important question that is potentially useful to clinical practice, and with wider social benefit to support mothers to breastfeed according to their goals and to recommended practice.
11. The justification for conducting such a study is well made in the introduction. It could be useful for the authors to reflect further based on their results, on what may distinguish the Japanese cultural context from other studies of this kind, and how those other socio-cultural contexts differs from Japan. It seems to me that the low paid labour force participation rate of Japanese new mothers is a rather unique cultural factor that could explain why practical help is less important than in other cultural contexts where early maternal return to paid employment is more common but unpaid housework and childcare still remains predominantly a female role. A sentence or two with citations and relevant data could strengthen the wider insights from the study.
12. The literature is clear that longer paid maternity leave is supportive of breastfeeding, and heavy unpaid work burdens involved in caring for an infant influence mothers’ infant feeding practices and ability to get sufficient rest and sleep etc. Japan is no different in that regard to other countries, and it would be interesting in a future study to survey and validate measures of key support requirements (and tradeoffs) involved in breastfeeding for those Japanese women who would prefer to maintain paid work participation, as well as breastfeed.

---

## Round 0.2 · accepted · Accept

Congratulations!

I want to express my sincere gratitude for submitting your valuable work to PeerJ.

·

Basic reporting

no comment - authors have addressed all my suggestions in the response/manuscript

Experimental design

no comment - authors have addressed all my suggestions in the response/manuscript

Validity of the findings

no comment - authors have addressed all my suggestions in the response/manuscript

Additional comments

Thank you to the authors for their clear responses. I have read through the response to authors and revised manuscript. I found the paper to be very clear and very interesting.

My only minor comment is that I am not sure the additional explanation of the statistical methods adds much to the paper, as they are quite basic methods and it may be more efficient to direct readers to textbooks/methods papers rather than outline the methods in more detail. However, I recognise this has been suggested by a reviewer, and it may be very useful for readers who are not familiar with the method, or those who hope to replicate/conduct a similar study.